# Does Music Therapy Improve Gait after Traumatic Brain Injury and Spinal Cord Injury? A Mini Systematic Review and Meta-Analysis

**DOI:** 10.3390/brainsci13030522

**Published:** 2023-03-21

**Authors:** Shashank Ghai

**Affiliations:** 1Psychology of Learning and Instruction, Department of Psychology, School of Science, Technische Universität Dresden, 01069 Dresden, Germany; shashank.ghai@tu-dresden.de; 2Centre for Tactile Internet with Human-in-the-Loop (CeTI), Technische Universität Dresden, 01069 Dresden, Germany

**Keywords:** music therapy, gait, neurorehabilitation, auditory cueing, traumatic brain injury, spinal cord injury

## Abstract

There is a growing body of research examining the potential benefits of music therapy-based auditory stimulation (MT) for individuals with movement disorders in improving gait performance. However, there is limited knowledge about the effects of MT on gait outcomes in individuals with traumatic brain injury (TBI) or spinal cord injury (SCI). A previous review of MT’s impact on gait in TBI had limitations, and there are no studies on its effects on gait in SCI. In this study, we conducted a meta-analysis to more thoroughly evaluate the impact of MT on gait outcomes in individuals with TBI and SCI. We systematically searched through eight databases and found six studies on MT in TBI and four on SCI. Our meta-analysis showed that MT has positive medium effect improvements on spatiotemporal aspects of gait in individuals with TBI (Hedge’s g: 0.52) and SCI (0.53). These findings suggest that MT could be a practical intervention for enhancing different aspects of gait in these populations, although the limited number and “fair” quality of the studies included in the meta-analysis may affect the generalizability of the outcomes. Further research is needed to fully understand the mechanisms by which MT may influence gait and determine the optimal parameters for its use.

## 1. Introduction

Traumatic brain injury (TBI) and spinal cord injury (SCI) are leading causes of disability worldwide [1], with gait disturbances being a common and potentially debilitating consequence [2,3,4]. Gait, or the manner in which an individual walks, is a complex motor task that requires the integration of sensory, cognitive, and motor processes [5]. These processes can be disrupted following TBI and SCI [2,4], leading to impairments that can impact mobility, independence, and quality of life [6,7]. Despite recent advancements in rehabilitation, gait deficits remain prevalent among individuals with TBI and SCI [8,9].

The use of music therapy-based auditory stimulations (MT) to achieve therapeutic goals has emerged as a promising intervention for individuals with TBI and SCI [10]. MT has been shown to influence various aspects of physical and cognitive functioning [11], including gait, in individuals with TBI and SCI [12,13,14]. Studies have suggested several methods by which MT can facilitate the spatial and temporal aspects of gait (i.e., speed, cadence, stride length etc.). For instance, the auditory stimulation during MT can provide external rhythmic cueing that can entrain or synchronize an individual’s gait pattern with the cue [15,16]. TBI can cause damage to various areas of the brain involved in timing and coordination, such as the cerebellum and basal ganglia [17,18], and as a result, these deficits in internal timing can manifest in various ways, including difficulty with gait, balance, and coordination [19]. Similarly, in SCI, depending upon the level and severity of the injury, the timing of movements, especially during gait, could be affected due to the upregulation of H-reflexes [20], loss of sensation [21], and muscle tone [22,23]. Here, auditory cueing with MT can provide an external, reliable source of temporal information that can be used to synchronize movement and improve gait stability, coordination, and efficiency [24,25,26]. Furthermore, MT can also be used to provide feedback on gait parameters and to distract individuals from pain or discomfort associated with gait [27,28,29].

Specifically, the neurophysiological mechanisms by which MT may improve gait in individuals with TBI and SCI are not fully understood [30,31], but may involve the activation of neural networks involved in gait [32], improvement in attention and executive function [33], and the modulation of emotional and behavioural responses [34,35,36,37]. Likewise, based on the existing evidence, it could be hypothesized that prolonged training with auditory stimulation could facilitate motor recovery in TBI and SCI by simply increasing connectivity between auditory and motor networks, especially in the alpha, beta, and gamma frequency bands [38,39]. In addition, MT may have aided in the recovery of gait in individuals with TBI and SCI, not only through its neurophysiological impact, but also by decreasing interference in cognitive-motor domains [40,41,42], enhancing joint proprioception [43,44,45], reducing variability in muscular co-activations [46], boosting motivation [47], and increasing arousal [48,49,50]. Furthermore, due to its dynamic character, MT has the ability to enhance motor performance in a patient-centred approach [51,52]. This approach involves mapping MT onto the individual’s movement characteristics, allowing it to adjust to the preferred cadence of the gait or the personalized movement characteristics of the performer [26,53]. This customization enables MT to be delivered according to the person’s preferences, such as being superimposed on their preferred type of music. This can lead to added benefits such as active participation and increased motivation for the performer [54,55,56]. These characteristics of MT align with the recommended best practice principles of neurorehabilitation, which advocate for interventions to be challenging, intensive, repetitive, intriguing, and highly task-specific in order to promote recovery [57,58,59]. 

Despite mounting evidence suggesting the beneficial influence of MT on spatiotemporal gait parameters, a lack of consensus exists in the literature regarding its efficacy. This lack of consensus exists primarily at the level of individual clinical trials. This lack of agreement is particularly evident at the level of individual clinical trials. For example, some trials have reported improvements in spatiotemporal outcomes of gait in TBI and SCI with the use of MT [16,60,61,62,63], whereas others have suggested that MT has either no effect [64,65], or that it can even adversely impact the gait parameters [66]. With respect to evidence-based synthesis, to date, only one meta-analysis study has assessed the influence of MT on gait performance in individuals with TBI [12], whereas one recent scoping review evaluated the influence of MT on SCI [13]. The meta-analysis for TBI has limitations in terms of both analytical and methodological aspects. Here, the meta-analysis included three studies reporting the outcome of gait speed, cadence, and stride length (i.e., a study by Nayak, Wheeler [67], Hurt, Rice [16], and a conference proceeding by Thaut and colleagues). The study by Nayak, Wheeler [67], which was included in the meta-analysis, had not even evaluated the influence of MT on spatiotemporal parameters of gait. Moreover, the review also cited a study by Thaut and colleagues, an abstract from a conference proceeding in which the authors presented the findings of Hurt and Rice [16] (i.e., the third study). This means that the authors Mishra, Florez-Perdomo [12] wrongly reported the outcomes of gait speed in their meta-analysis. This discrepancy in terms of the overall meta-analysis raises questions regarding the scientific vigour of their analysis and, eventually, their overall findings. Regarding the scoping review among individuals with SCI, a lack of statistical analysis concerning the influence of MT on gait outcomes limits our ability to interpret the results regarding the overall magnitude of impact MT has on gait recovery following SCI. 

In light of the current gaps in the literature, this mini-review aims to systematically evaluate the impact of MT on various spatiotemporal parameters of gait, including gait speed, cadence, stride length, and step length, on individuals with TBI and SCI through a meta-analytic approach. Additionally, this review will examine the effect of MT on gait symmetry in individuals with TBI.

## 2. Materials and Methods

The systematic review and meta-analysis were performed in accordance with the PRISMA-SR 2020 guidelines. The checklist for this process is included in Appendix A. The review had been pre-registered on the Open Science Framework (https://osf.io/crmpw).

### 2.1. Data Sources and Search Strategy

The systematic literature search was conducted across eight databases (EMBASE, PROQUEST, Psychinfo, PEDro, Web of Science, Pubmed, EBSCO, Scopus) for the publication period from January 1970 until January 2023. These databases were chosen on the basis of access provided by the academic organization. The appropriate PICOS search terms have been provided in Appendix A. The authors also searched the reference section of the included studies.

The search criteria for selecting appropriate studies in the review were developed according to the PICOS approach (Population, Intervention, Comparator, Outcome of interest, and Study design). The criteria for inclusion were developed by two authors (S.G, I.G). A detailed list of relevant search terms used has been provided in the pre-registration protocol. The inclusion criteria were as follows: (1) Population groups with TBI; (2) Population groups with SCI; (3) Studies assessing the effect of MT on spatiotemporal parameters of gait; (4) Studies assessing the effect of MT on gait symmetry; (5) All types of quantitative clinical studies including randomized controlled trials, controlled clinical trials, crossover trials, longitudinal studies, cohort analyses, case series, feasibility studies, and case studies will be included; (6) Studies scoring more than or equal to 4 on the PEDro scale; (7) Studies published in peer-reviewed academic journals; and (8) Studies published in either English, French, German, or the Hindi language. 

The screening of the titles, abstracts and full texts of all the studies was conducted by two authors. In the case of discrepancies regarding the selection of relevant studies, discussions were held between the two authors. The following information was extracted from the articles: name of authors, country, demographic information (i.e., participant age, total sample size, sex), Glasgow Coma Scale information for individuals with TBI, ASIA scale information for individuals with TBI, years since injury, assessed outcomes, MT training schedule, MT characteristics, and the result of the studies. 

### 2.2. Evaluation of the Methodological Quality

The methodological quality of the included studies utilized the PEDro scale [68]. The PEDro scoring quality appraisal can be interpreted as follows: studies scoring between 9 to 11 are considered of “excellent quality”, 6 to 8 are of “good quality”, 4 to 5 are of “fair quality”, and those with a score of less than or equal to 3 are considered to be of “poor quality” [69]. The appraisal of the studies was conducted by two authors independently.

### 2.3. Data Analysis

A random effect meta-analysis was conducted with Comprehensive meta-analysis (V 4.0) [70]. We carried out within-group analyses using the respective studies’ spatiotemporal gait parameters and gait symmetry. A between-group meta-analysis was not conducted due to the paucity of data concerning the control group in the included studies. The meta-analysis results included weighted and adjusted effect size (i.e., Hedge’s g), 95% confidence interval (C.I.), and significance level. The effect size was interpreted as small for <0.16, medium for ≥0.38 to 0.76, and large for >0.76 [71]. The results were presented in forest plots. The heterogeneity of the included studies was quantified using I^2^ statistics. Heterogeneity was considered negligible for 0% to 25%, moderate for 25% to 75%, and substantial for >75% [72]. We also conducted “leave-one-out” sensitivity analyses to test the robustness of our findings and explore the heterogeneity. The method systematically removes each study from the meta-analysis and re-analyzes the data to assess the influence of individual studies on the overall results. This helps to identify studies that may be driving the results and assess the robustness of the findings [73]. Additionally, an assessment of publication bias was carried out according to the trim and fill procedure by Duval and Tweedie [74]. The study’s significance level was set at 5%.

## 3. Results

After searching through nine databases and one registry, a total of 2356 articles were found. The articles were then screened using the PICOS inclusion criteria, resulting in only 10 articles being included. The entire selection process is illustrated in Figure 1 [75]. The qualitative data were then extracted from all of the included studies, as shown in Table 1 and Table 2. The data from one case study could not be included in the meta-analysis because the data of only a single participant was reported in the study [76].

### 3.1. Study Design

Of the ten included studies, one was a randomized controlled trial [61], six were quasi-experimental studies [16,60,62,63,64,66], two were case studies [65,77], and one was a case study [76].

### 3.2. Country of Research

Six of the included studies were conducted in the USA [16,62,63,65,66,76], and one each was conducted in India [61], Italy [64], South Korea [77], and Thailand [60]. 

### 3.3. Risk of Bias

The individual PEDro scoring of each included study is presented in Figure 2 and Table 3. In the included studies, two studies scored 6 [61,66], five studies scored 5 [16,60,62,63,64], and three studies scored 4 [64,76,77]. The included studies had an average PEDro score of 4.9 ± 0.7, indicating a “fair” overall quality of the studies.

### 3.4. Publication Bias

Figure 3 demonstrates the occurrence of publication bias using Duval and Tweedie’s trim and fill procedure. The results showed no indication of missing studies on either side of the mean effect. The combined studies were analyzed using the random effect model, and the point estimate and 95% confidence interval (C.I.) were 0.58 and 0.28 to 0.88, respectively. The use of the trim and fill procedure did not alter these values.

### 3.5. Systematic Review Report

#### 3.5.1. Participants

The data from a total of 31 (11F, 20M) individuals with TBI and 58 (11F, 47M) individuals with SCI were reported in the included studies. The average age of the individuals with TBI was (31.3 ± 11.9 years), whereas the average age for individuals with SCI was (38.1 ± 5.3 years). 

#### 3.5.2. Years since Injury

Seven studies had reported the information concerning years since injury for individuals with TBI [16,63,76,77] and SCI [60,64,66]. Three studies had not reported the information concerning the years since injury [61,62,65]. The range for years since injury for the cohort with TBI was 0.3 to 16.9 years. The range of years since injury for individuals with SCI was 0.3 years to 27 years.

#### 3.5.3. Outcome

According to the qualitative evidence gathered in the current review, MT appears to have a positive impact on spatiotemporal parameters of gait among individuals with TBI and SCI. More precisely, five studies focusing on individuals with TBI reported an improvement in spatiotemporal outcomes of gait following MT [16,62,63,76,77]. One case series reported improvement in the spatiotemporal outcomes for one of their participants, whereas deterioration in the gait outcomes was reported for the other participant [65]. Concerning the individuals with SCI, MT was reported to improve spatiotemporal outcomes of gait in two studies [60,61], whereas one study reported no difference [64], and one reported a deterioration in gait performance [66]. 

#### 3.5.4. Characteristics of Music Therapy

Different variations of MT were used in the included studies (see Table 1 and Table 2). In the studies evaluating the influence of MT among individuals with TBI, three studies provided rhythmic stimulations at the preferred cadence of their cohort [16,63,77]. Two studies provided rhythmic stimulation at a predetermined frequency [62,65], and one provided rhythmic stimulation during rhythmic exercises [76]. Similarly, for individuals with SCI, the rhythmic stimulations were delivered as per the preferred cadence of the individuals by two studies [61,66]. One study reported that they delivered rhythmic stimulation at a 25% faster pace than their cohort’s preferred cadence [60], and one study delivered MT based on the load their cohort imparted on their crutch [64]. 

In terms of the acoustic signal characteristics for studies in TBI, three studies delivered rhythmic stimulation with a metronome [16,62,76], and three studies used rhythmic stimuli superimposed on music [63,65,77]. Concerning studies in SCI, two studies used a metronome [60,61], one study mentioned that they used either a metronome or rhythmic stimulation delivered with synthesized guitar [66], and one study provided high and low pitch tones based on the load imparted on crutches [64].

### 3.6. Meta-Analysis Report

A detailed report of the within-group meta-analysis can be found in Table 4 and Table 5.

#### Sensitivity Analysis

A comprehensive account of the leave-one-out sensitivity analysis is presented in Table 6. In particular, studies were reported in the table if the significance level of the global analysis was less than 0.05 and the exclusion of any individual study caused the significance level to rise above this threshold. Conversely, studies were also reported if the overall analysis was not significant at a 0.05 level, and the exclusion of any specific study led to a decrease in the significance level below this threshold.

## 4. Discussion

The primary objective of this systematic review and meta-analysis was to consolidate the existing knowledge about the effects of MT on spatiotemporal parameters of gait in individuals with TBI and SCI. Based on the results of the exploratory meta-analysis, there appears to be a substantial impact of MT on all spatiotemporal gait outcomes in people with TBI. However, concerning SCI, while the enhancements in spatiotemporal parameters of gait were *medium* to *large* in magnitude, they were not statistically significant.

To date, a single meta-analysis has quantitatively evaluated the impact of MT on spatiotemporal parameters of gait in individuals with TBI. The review, which included three studies, reported a statistically significant enhancement in stride length (*p* = 0.0007), but no significant improvement in gait velocity (12.2 cm/sec) or cadence (7.19 steps/minute) with the use of MT. However, it should be noted that the findings of this study should be viewed with caution due to the inclusion of a study in the review that did not evaluate gait velocity as an outcome [67], and the inclusion of a conference proceeding that presented data already published in another study by Hurt and Rice [16]. This was confirmed by the author (S.G) in correspondence with Professor Michael Thaut. Additionally, we encountered a thorough scoping review, which evaluated the impact of motor training on gait outcomes in individuals with SCI [13]. The review qualitatively identified three studies that examined the influence of motor training on gait outcomes in the SCI population. However, it should be noted that the results of this scoping review do not provide statistical evidence of the magnitude of the effect of motor training on spatiotemporal gait parameters. 

In the present study, we aimed to expand upon the findings of previous studies by including a higher number of studies and conducting a comprehensive meta-analysis for both individuals with TBI and SCI. Consistent with previous literature, where spatiotemporal gait parameters serve as a means of quantifying both short-term and training-related alterations in gait speed [78,79], the results indicated that MT led to a significant *medium* effect enhancement in gait speed (Hedge’s g: 0.64, *p* = 0.04), cadence (0.49, *p* = 0.04), and stride length (0.73, *p* = 0.02) for individuals with TBI. We also conducted an analysis to quantify the effect of MT on step length parameters. Our results demonstrated a small non-significant enhancement in step length (0.19, *p* = 0.51). Similarly, in the case of SCI, we observed a non-significant *small*-to-*large* effect enhancement in gait speed (0.76, *p* = 0.37) and cadence (0.22, *p* = 0.26). We presume that the variation in the magnitude of improvement observed in the spatiotemporal parameters of gait may be attributed to several factors. Firstly, the limited number of studies included in the meta-analyses (i.e., gait speed: five, cadence: five, stride length: three, step length: three) may have reduced the statistical power and increased the variability of the results. Secondly, there was a marked discrepancy in the designs of the studies included in the analysis, specifically in the use of quasi-experimental [16,60,62,63,66] and case series designs [65,77]. Furthermore, the sample size in the analyses of spatiotemporal gait parameters was relatively small (i.e., for TBI gait speed: 31 participants, stride length: 18, step length: 15; for SCI gait speed: 54, cadence: 50), which may have further contributed to the observed differences in gait outcomes. Thirdly, the variability in gait-related impairments due to the broad nature of TBI and SCI can result in different responses to interventions among individuals [16,62,65,80]. Hurt and Rice [16] proposed that the high level of variability observed in individuals following TBI may be attributed to diffuse axonal injuries resulting in injury to sub-cortical white matter structures, such as the corpus callosum and superior cerebellar peduncles. Additionally, the authors posited that damage to the temporal and frontal lobes, as well as the midbrain, may also contribute to limitations in the ability to process auditory-motor information by impacting regions such as the motor cortex, the pre-motor cortex, and the auditory cortex [16]. Based on this evidence, we hypothesize that this high level of inter-individual variability may have played a role in the differences observed in overall spatiotemporal gait outcomes. Lastly, it is possible that the moderate increase in cadence and minimal increase in step length may have resulted in a significant improvement in gait speed. In the context of individuals with SCI, the limited number of studies evaluating the effect of muscle training on the spatial aspect of gait [61], such as stride length or step length, precluded the confirmation of the aforementioned effect.

There are multiple mechanisms in the literature that could account for the improvements in gait performance observed in individuals with TBI and SCI. Some studies suggest that the primary mechanism behind these enhancements is the ability of MT to promote task-specific, challenging, motivating, immersive, and multisensory learning [31,48,49,81]. This is particularly relevant for individuals with TBI and SCI, who often face challenges in their sensory domains, such as audition and proprioception, which limit their ability to learn and perform motor tasks [82,83]. Thompson and Hays [63] proposed that the rhythmic stimulation during MT may have activated the auditory-motor networks, leading to auditory-motor synchronization in their cohort of individuals with TBI who were outside the typical window of natural neurological recovery. The authors also reported that the MT intervention was well-tolerated by participants, as none of them experienced any adverse events or falls during the study. Furthermore, the authors observed that in addition to the improvements in spatiotemporal gait parameters, their participants also demonstrated almost clinically meaningful improvements in Functional Gait Assessment scores after the MT follow-up (i.e., follow-up vs. pre-intervention: 17.8 vs. 14.2) [84]. Similarly, Wilfong [62] reported an improvement in spatiotemporal gait performance, including an increase in speed (13.2%), cadence (6.6%), and stride length (10.7%) in their study sample. The authors attributed these enhancements to the ability of a rhythmic tempo to effectively manage muscle timing during gross movement tasks. In the population of individuals with SCI, previous research has demonstrated that the use of external rhythmic entrainment in combination with MT can bypass deficits in the internal referencing system for movement correction in the somatosensory cortex, thereby facilitating internal dynamics through the entrainment of phase-related coupling among body segments [60,85]. This study also reported that the implementation of MT led to participants walking faster (i.e., 0.43 vs. 0.40 m/sec) than their own self-determined faster pace, suggesting that external information can enable individuals to surpass their own perceived capabilities [60]. Additionally, in individuals with SCI, disruptions in descending control of central pattern generators (CPGs) may result in impairments in gait performance [86]. To address this issue, Singhal and Kataria [61] proposed the use of external cueing with MT as a means of influencing CPGs via the basal ganglia and lower brainstem reticulospinal neurons. The authors suggest that external cueing provided by MT may enable CPGs in SCI individuals to adapt motor patterns during gait to the external stimuli. This hypothesis is based on the principle that auditory cueing can entrain or synchronize neural oscillations within CPGs, thereby influencing the timing and coordination of motor output.

Furthermore, it is well-established that individuals with TBI and SCI frequently exhibit increased gait asymmetry, which is a prevalent and persistent deficit [16,87,88]. This asymmetric nature of gait can further exacerbate the high levels of variability in gait and increase the risk of falls in these individuals [88,89]. In a case report by Sheridan and Thaut [65], it was found that MT led to reduced variability in the step time and step length parameters in one of the participants, in addition to improvements in walking endurance, community balance, and mobility. Furthermore, the authors suggested that MT not only improves dynamic balance and mobility after TBI, but also facilitates community integration. Similar findings have been reported by Hurt and Rice [16], who reported increased stride symmetry with MT during both normal and fast-paced gait, and suggested that the increased symmetry could indicate rhythmic entrainment to the temporal beat symmetry. Amatachaya and Keawsutthi [60] also found that in their cohort of SCI, auditory feedback (91.5%) from MT led to the highest step symmetry compared to visual feedback (86.5%) or no feedback (82.8%). In a meta-analysis, a large magnitude increase in gait symmetry was reported for individuals with TBI (1.28), while outcomes regarding gait symmetry in individuals with SCI were not reported due to a lack of data.

### 4.1. Limitations

While the objective of the study was to investigate the effect of MT on spatiotemporal parameters of gait in individuals with TBI and SCI, the included studies varied in their assessment of MT and MT-based training on gait outcomes. Subgroup analyses were conducted to differentiate the effects of MT-based training and simple MT, but discrepancies in the included studies still remained. For instance, the varying duration of MT-based training among studies made it difficult to determine the most effective training dosage. Moreover, the study’s findings regarding the influence of MT on SCI are limited because the review only included studies with individuals classified as ASIA C or D. These individuals had neurological damage but were still able to rehabilitate their gait. In contrast, individuals with type B and A lesions, who typically cannot move, were not assessed. Therefore, it is important to interpret the study’s results with caution, as they may not be applicable to the entire SCI population.

In addition, the studies included in this review also varied regarding the implementation of MT. Here, while some studies offered MT at the participant’s preferred cadence [16,61,66], others did not [65,76]. There were also variations in the characteristics of the auditory signals used, including embedding the stimulus in music [63,77], or using a simple metronome [62]. These differences highlight the importance of categorizing MT-based interventions based on auditory signal characteristics, training dosages, and their relevance to rehabilitation in future studies. Another limitation of this study was the inclusion of studies with small sample sizes, such as case series and case studies. This could have influenced the results as small sample size studies are known to produce high variability in the results, reduce the power of the analysis, and increase the likelihood of type II errors. However, the reason for including these studies was that, due to the lack of large-scale studies in the current literature, the aim was to include as many studies as possible to provide an overview of the influence of MT on gait outcomes in TBI and SCI. Another limitation of our review is that we did not include studies that investigated the effects of MT on cognitive and psychological outcomes in individuals with TBI and SCI, as they fell outside the scope of our research question. Although several studies have reported the positive effects of MT on these outcomes [90,91,92], we were unable to evaluate them in our review. However, we suggest that future systematic reviews be conducted to establish the current state of evidence regarding the impact of MT on cognitive and psychological outcomes in individuals with TBI and SCI. Despite these limitations, the current study provides important information that could contribute to the development of more effective rehabilitation strategies for individuals with TBI and SCI.

### 4.2. Future Directions

The current literature on the utilization of MT for gait rehabilitation in individuals with TBI and SCI is limited in comparison to other neurological conditions such as Parkinson’s disease [26,93], stroke [94,95], and cerebral palsy [96]. Thus, it is crucial for future research to investigate the potential benefits of MT on gait outcomes in TBI and SCI populations. Such findings have the potential to significantly improve the rehabilitation outcomes for individuals with debilitating gait deficits and provide clinical professionals with valuable information for incorporating MT into the gait rehabilitation of TBI and SCI patients. Additionally, beyond evaluating the effects of conventional rhythmic MT on gait outcomes in TBI and SCI patients, we suggest that future studies explore the use of concurrent MT interventions. Movement sonification is one such approach, and it involves transforming kinematic movement parameters into real-time auditory signals that provide feedback stimuli, potentially enhancing motor perception and performance by targeting neural networks involved in biological motion perception. Previous neuroimaging studies have demonstrated that passively listening to sonified human actions that are congruent with the observed movement can enhance movement timing and auditory-motor entrainment effects [97]. This may be due to the close relationship between the stimuli and biological motion, which activates the human action observation network [98]. Moreover, behavioral studies have indicated that sonification can enhance proprioceptive accuracy and assist in synchronizing cyclic movement patterns [44,45]. Therefore, training with movement sonification may help individuals with TBI and SCI better perceive their own movement patterns and determine optimal movement amplitudes for effective gait performance.

## 5. Conclusions

To summarize, the meta-analysis concludes that MT has a positive impact on spatiotemporal parameters of gait in individuals with TBI and SCI. In individuals with TBI, MT led to improvements in gait speed, cadence, stride length, step length, and gait symmetry, while in individuals with SCI, it led to improvements in gait speed and cadence. However, it is important to note that the studies included in the analysis were of “fair” methodological quality and had a smaller sample size. Although sensitivity analyses confirmed the robustness of the overall findings, the removal of some studies affected the *p*-value and overall effect size for gait speed and cadence in both TBI and SCI. As such, the findings should be interpreted with caution. To establish more reliable, evidence-based guidelines for the use of MT in gait rehabilitation after TBI and SCI, future high-quality trials are recommended to further evaluate its influence on gait outcomes.

## Figures and Tables

**Figure 1 brainsci-13-00522-f001:**
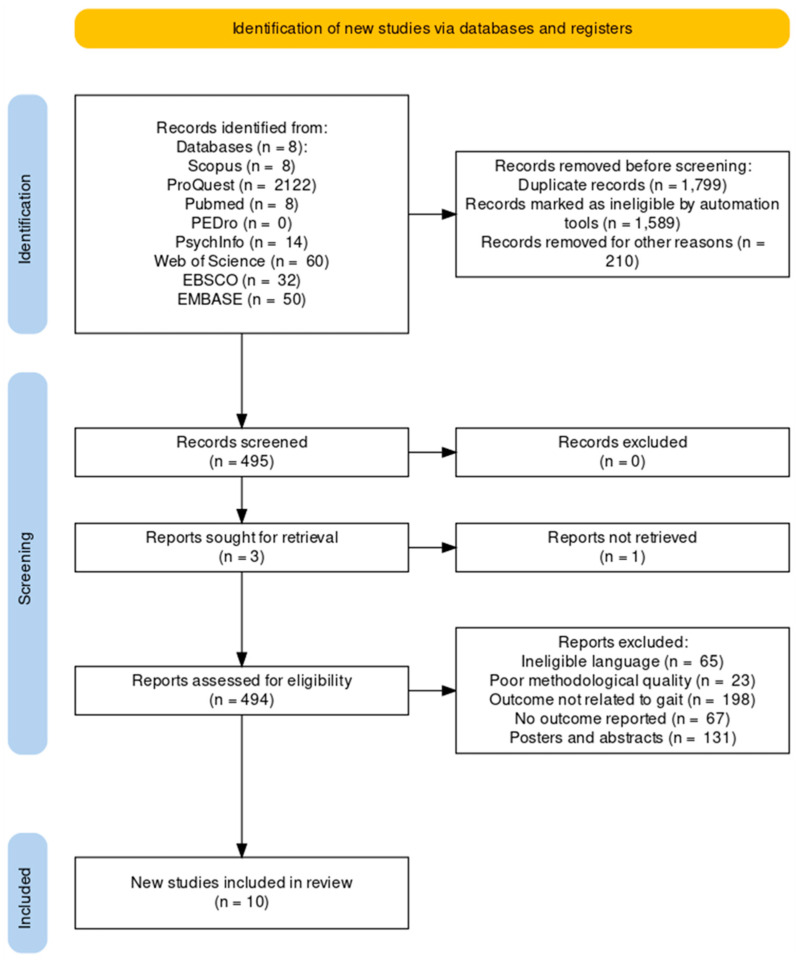
PRISMA flowchart.

**Figure 2 brainsci-13-00522-f002:**
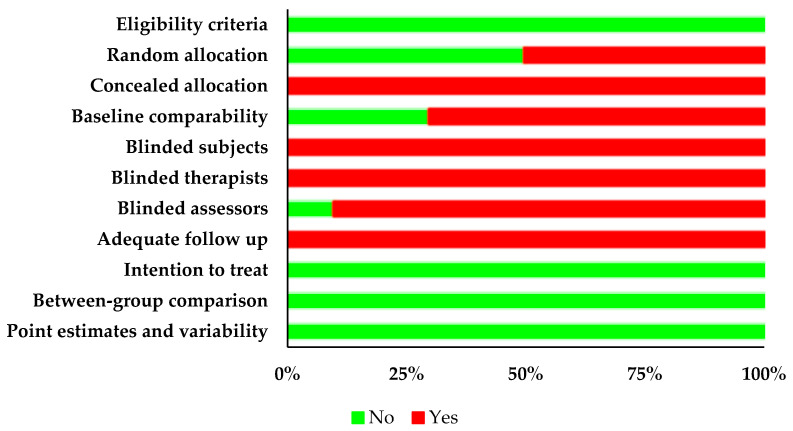
Risk of bias based on the PEDro scale (“No” indicates the absence of risk of bias and “Yes” indicates the presence of risk of bias).

**Figure 3 brainsci-13-00522-f003:**
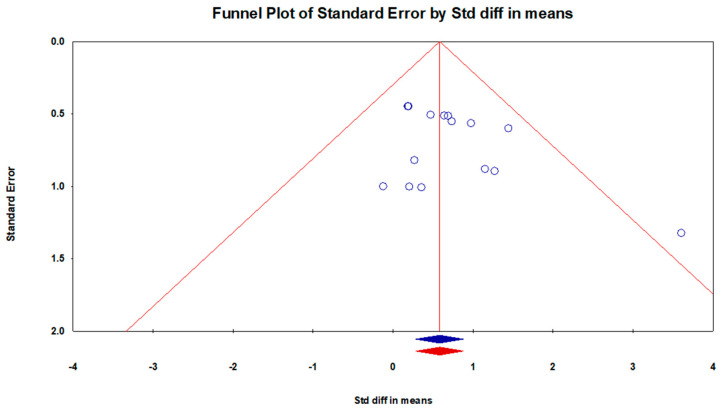
Duval and Tweedie’s trim and fill procedure. The blue circles represent individual studies, while the funnel plot encompasses 95% of the pseudo-confidence intervals, and the vertical midline indicates the estimated overall effect size, which includes both observed and imputed studies.

**Figure 4 brainsci-13-00522-f004:**
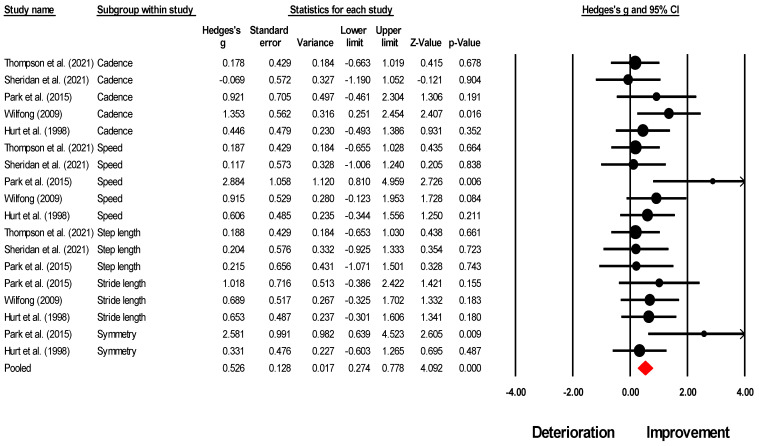
A forest plot depicts the impact of MT on overall spatiotemporal gait outcomes in individuals with TBI. It includes individual weighted effect size Hedge’s g represented as black circles, and the whiskers represent the 95% confidence intervals. The pooled weighted effect size and 95% CI are presented at the bottom with a red diamond. A positive overall effect size in this analysis implies an enhancement in spatiotemporal outcomes of gait with MT, while a negative overall effect indicates a decline in spatiotemporal outcomes of gait with MT. Refs [16,62,63,65,77] mentioned.

**Figure 5 brainsci-13-00522-f005:**
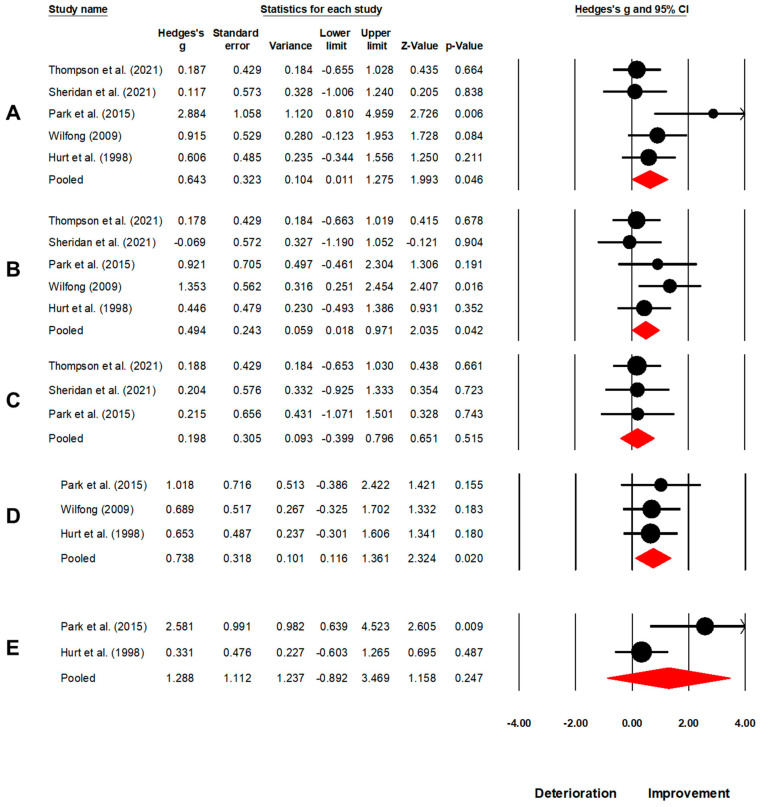
A forest plot depicts the impact of MT on (**A**) gait speed, (**B**) cadence, (**C**) step length, (**D**) stride length, and (**E**) gait symmetry in individuals with TBI. It includes individual weighted effect size Hedge’s g represented as black circles, and the whiskers represent the 95% confidence intervals. The pooled weighted effect size and 95% CI are presented at the bottom with a red diamond. A positive overall effect size in this analysis implies an enhancement in spatiotemporal outcomes of gait with MT, while a negative overall effect indicates a decline in spatiotemporal outcomes of gait with MT. Refs [16,62,63,65,77] mentioned.

**Figure 6 brainsci-13-00522-f006:**
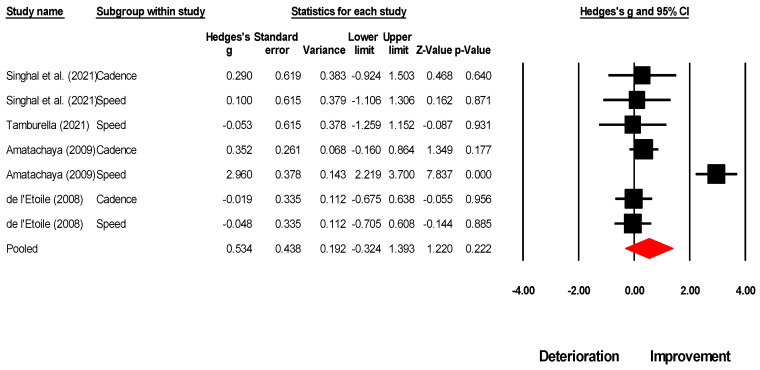
A forest plot depicts the impact of MT on overall spatiotemporal gait outcomes in individuals with SCI. It includes individual weighted effect size Hedge’s g represented as black circles, and the whiskers represent the 95% confidence intervals. The pooled weighted effect size and 95% CI are presented at the bottom with a red diamond. A positive overall effect size in this analysis implies an enhancement in spatiotemporal outcomes of gait with MT, while a negative overall effect indicates a decline in spatiotemporal outcomes of gait with MT. Refs [60,61,64,66] mentioned.

**Figure 7 brainsci-13-00522-f007:**
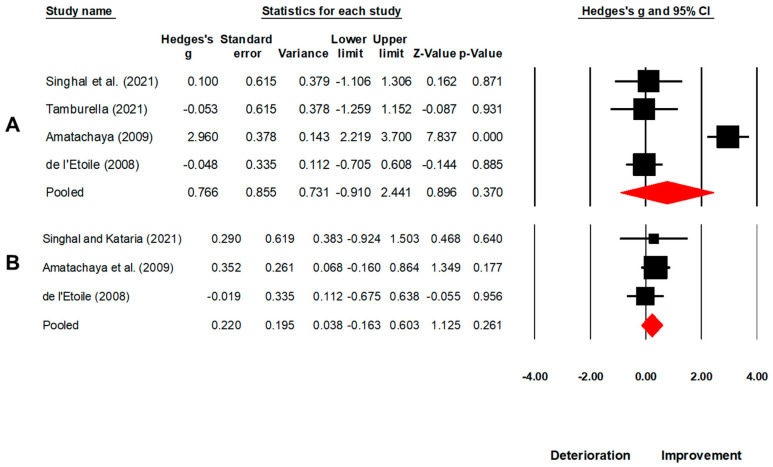
A forest plot depicts the impact of MT on (**A**) gait speed and (**B**) cadence in individuals with SCI. It includes individual weighted effect size Hedge’s g represented as black circles, and the whiskers represent the 95% confidence intervals. The pooled weighted effect size and 95% CI are presented at the bottom with a red diamond. A positive overall effect size in this analysis implies an enhancement in spatiotemporal outcomes of gait with MT, while a negative overall effect indicates a decline in spatiotemporal outcomes of gait with MT. Refs [60,61,64,66] mentioned.

**Table 1 brainsci-13-00522-t001:** Details of studies evaluating individuals with traumatic brain injury.

Authors Country of Research	Sample Size (N) Gender Distribution (F, M) (Age in Years as Mean ± SD/Range)	Glasgow Coma Scale Years Since Injury	Outcomes	Training Schedule	Music Therapy (MT) Characteristics	Results
Thompson, Hays [63] USA	N = 10 2F, 8M (37.9 ± 15.2)	4.1 ± 1.6 1.3 to 16.9	Gait speed Cadence Step length Functional gait assessment 10-m walk test (meter/sec) 10-m walk test (sec)	Session length: 30 min Times per week: - Weeks: 2 Total sessions: 10	Rhythmic click as per preferred cadence added to preferred music	Gait speed: ↑ with MT. Cadence: ↑ with MT. Step length: ↑ with MT. Functional gait assessment: ↑ with MT. 10-m walk test (meter/sec): ↑ with MT. 10-m walk test (sec): ↓ with MT.
Sheridan, Thaut [65] USA	N = 1 1M 42	- -	Preferred pace, maximum pace Gait speed Cadence Step length Step time variability Step length variability Step width variability Clinical gait and balance measures 6-min walk test	Session length: 30 min Times per week: 3 Weeks: 3	Rhythmic auditory stimulation with music recordings at a predetermined frequency	Gait speed: ↑ with MT. Cadence: No difference. Step length: ↑ with MT. Step time variability: ↓ with MT. Step length variability: ↓ with MT. Step width variability: No difference.
	N = 1 1M 54					Gait speed: ↓ with MT. Cadence: ↓ with MT. Step length: ↓ with MT. Step time variability: ↑ with MT. Step length variability: ↑ with MT. Step width variability: ↑ with MT.
Park [77] South Korea	N = 1 1M (10)	- 0.6	Gait speed Cadence Step length Stride length Step time Stride time Gait symmetry	Session length: 30 min Times per week: - Weeks: 3 Total sessions: 8	Rhythmic harmonic stimulation at preferred cadence with music	Gait speed: ↑ with MT. Cadence: ↑ with MT. Step length: ↑ with MT on the left side, ↓ with MT. On the right side. Stride length: ↑ with MT. Step time: ↓ with MT on the left side, ↑ with MT. On the right side. Stride time: ↑ with MT. Gait symmetry (kinematic parameters of hip and knee): ↑ with MT.
	N = 1 1F (14)	- 0.6				Gait speed: ↑ with MT. Cadence: ↑ with MT. Step length: ↑ with MT. Stride length: ↑ with MT. Step time: ↑ with MT. Stride time: ↑ with MT. Gait symmetry (kinematic parameters of hip and knee): ↑ with MT.
	N = 1 1M (16)	- 1.1				Gait speed: ↑ with MT. Cadence: ↑ with MT. Step length: ↑ with MT. Stride length: ↑ with MT. Step time: ↑ with MT. Stride time: ↑ with MT. Gait symmetry (kinematic parameters of hip and knee): ↑ with MT.
Goldshtrom, Knorr [76] USA	N = 1 1F 24	- 9	Gait speed Cadence	Session length: - Times per week: - Weeks: -	Rhythmic exercise program with auditory cues	Gait speed: ↑ with MT. Cadence: ↑ with MT.
Wilfong [62] USA	N = 7 3F, 4M (34.7 ± 13.6)	- -	Gait speed Cadence Stride length	Session length: 15 min Times per week: 3 Weeks: 3	Rhythmic auditory stimulation with a timed metronome	Gait speed: ↑ with MT. Cadence: ↑ with MT. Stride length: ↑ with MT.
Hurt, Rice [16] USA	N = 8 3F, 5M (30 ± 5)	- 0.3 to 2	Normal gait, fast gait Gait speed Cadence Stride length Gait symmetry	Session length: - Times per week: - Weeks: -	Rhythmic auditory stimulation at the preferred cadence	Normal gait Gait speed: ↑ with MT. Cadence: ↑ with MT. Stride length: ↑ with MT. Gait symmetry: ↑ with MT. Fast gait Gait speed: ↓ with MT. Cadence: ↓ with MT. Stride length: ↓ with MT. Gait symmetry: ↑ with MT.
				Session length: - Times per week: 7 Weeks: 5		Normal gait Gait speed: ↑ with MT. Cadence: ↑ with MT. Stride length: ↑ with MT. Gait symmetry: ↑ with MT. Fast gait Gait speed: ↑ with MT. Cadence: ↑ with MT. Stride length: ↑ with MT. Gait symmetry: ↑ with MT.

F: Female, M: Male, MT: Music therapy.

**Table 2 brainsci-13-00522-t002:** Details of studies evaluating individuals with spinal cord injury.

Authors Country of Research	Sample Size (N) Gender Distribution (F, M) (Age in Years as Mean ± SD/Range)	ASIA Score Years Since Injury	Outcomes	Training Schedule	Music Therapy (MT) Characteristics	Results
Singhal and Kataria [61] India	MT: N = 4 4M (32.2 ± 16.8)	ASIA C: 2 ASIA D: 2 -	Gait speed Cadence Step length Walking index for spinal cord injury II	Session length: 30 min Times per week: - Weeks: 2 Total sessions: 10	Rhythmic auditory stimulation at preferred cadence with a metronome with bodyweight supported treadmill	Gait speed: ↑ with MT. Cadence: ↑ with MT. Step length: ↑ with MT. Walking index for spinal cord injury II: ↑ with MT.
	Ct: N = 4 4M (32 ± 4)	ASIA C: 2 ASIA D: 2 -			Bodyweight supported treadmill	Gait speed: ↑ with MT. Cadence: No difference. Step length: ↑ with MT. Walking index for spinal cord injury II: ↑ with MT.
Tamburella, Lorusso [64] Italy	N = 4 4M (35.2 ± 15.5)	ASIA D: 3 One patient not specified 0.30 to 1	Gait speed	Session length: - Times per week: - Weeks: - Total sessions: 1	Load-related auditory feedback (low and high pitch tones) with a crutch	Gait speed: No difference.
Amatachaya, Keawsutthi [60] Thailand	N = 29 7F, 22M (44 ± 15.2)	ASIA C: 4 ASIA D: 25 16 to 27	Gait speed Stride length Cadence Step symmetry	Session length: - Times per week: - Weeks: -	Rhythmic auditory stimulation with metronome 25% faster than preferred cadence	Gait speed: ↑ with MT. Stride length: No difference. Cadence: ↑ with MT. Step symmetry: ↑ with MT.
de l’Etoile [66] USA	N = 17 4F, 13M (41)	- 5.8 ± 4.8	Gait speed Cadence Stride length	Session length: - Times per week: - Weeks: -	Rhythmic auditory stimulation at the preferred cadence	Gait speed: ↓ with MT. Cadence: ↓ with MT. Stride length: ↑ with MT.
					Rhythmic auditory stimulation at 5% faster than normal cadence	Gait speed: ↓ with MT. Cadence: ↓ with MT. Stride length: ↓ with MT.

F: Female, M: Male, ASIA: American Spinal Injury Association classification, MT: Music therapy.

**Table 3 brainsci-13-00522-t003:** Detailed PEDro scoring (+: bias absent, -: bias present).

	Overall Score	Point Estimates and Variability	Random Allocation	Between-Group Comparison	Intention to Treat	Blinded Subjects	Adequate Follow-Up	Blinded Assessors	Blinded Therapists	Baseline Comparability	Concealed Allocation	Eligibility Criteria
Singhal and Kataria [61]	6	+	+	+	+	-	-	-	-	+	-	+
Tamburella, Lorusso [64]	5	+	-	+	+	-	-	-	-	+	-	+
Thompson, Hays [63]	5	+	-	+	+	-	-	-	-	+	-	+
Sheridan, Thaut [65]	4	+	-	+	+	-	-	-	-	-	-	+
Park [77]	4	+	-	+	+	-	-	-	-	-	-	+
Goldshtrom, Knorr [76]	4	+	-	+	+	-	-	-	-	-	-	+
Amatachaya, Keawsutthi [60]	5	+	+	+	+	-	-	-	-	-	-	+
Wilfong [62]	5	+	+	+	+	-	-	-	-	-	-	+
de l’Etoile [66]	6	+	+	+	+	-	-	+	-	-	-	+
Hurt, Rice [16]	5	+	+	+	+	-	-	-	-	-	-	+

**Table 4 brainsci-13-00522-t004:** Meta-analysis outcome.

Number	Outcome	Number of Studies Included in the Analysis; (References)	Meta-Analysis Result Hedge’s g, 95% C.I., *p*-Value	Heterogeneity I^2^ Stastistics	Figure Number
1.	Overall spatiotemporal outcomes	N = 5; [16,62,63,65,77]	0.52, 0.27 to 0.77, *p* < 0.001	1%	Figure 4
2.	Gait speed	N = 5; [16,62,63,65,77]	0.64, 0.01 to 1.27, *p* = 0.046	40%	Figure 5A
3.	Cadence	N = 5; [16,62,63,65,77]	0.49, 0.01 to 0.97, *p* = 0.042	5%	Figure 5B
4.	Step length	N = 3; [63,65,77]	0.19, −0.40 to 0.79, *p* = 0.515	0%	Figure 5C
5.	Stride length	N = 3; [16,62,77]	0.73, 0.11 to 1.36, *p* = 0.020	0%	Figure 5D
6.	Gait symmetry	N = 2; [16,77]	1.28, −0.89 to 3.46, *p* = 0.247	0%	Figure 5E

**Table 5 brainsci-13-00522-t005:** Meta-analysis outcome for spinal cord injury.

Number	Outcome	Number of Studies Included in the Analysis; (References)	Meta-Analysis Result Hedge’s g, 95% C.I., *p*-Value	Heterogeneity I^2^ Stastistics	Figure Number
1.	Overall spatiotemporal outcomes	N = 4; [60,61,64,66]	0.534, −0.32 to 1.39, *p* = 0.222	88%	Figure 6
2.	Gait speed	N = 4; [60,61,64,66]	0.76, −0.91 to 2.44, *p* = 0.370	93%	Figure 7A
3.	Cadence	N = 3; [60,61,66]	0.22, −0.16 to 0.60, *p* = 0.260	0%	Figure 7B
4.	Step length	N = 1; [61]	-	-	-
5.	Stride length	-	-	-	-
6.	Gait symmetry	-	-	-	-

**Table 6 brainsci-13-00522-t006:** Leave one out sensitivity analysis.

Number	Analysis	Meta-Analysis *p*-Value	I^2^	Studies Impacting *p*-Value upon Removal	*p*-Value upon Removal	Figure
Traumatic brain injury						
1.	Overall spatiotemporal outcomes	<0.001	1%	No effect	-	Appendix A
2.	Gait speed	0.046	40%	Park [77] Wilfong [62] Hurt, Rice [16]	0.074 0.139 0.104	Appendix A
3.	Cadence	0.042	5%	Park [77] Wilfong [62] Hurt, Rice [16]	0.116 0.238 0.103	Appendix A
4.	Step length	0.515	0%	-	-	-
5.	Stride length	0.020	0%	-	-	-
6.	Gait symmetry	0.247	0%	-	-	-
Spinal cord injury						
7.	Overall spatiotemporal outcomes	0.222	88%	No effect	-	Appendix A
8.	Gait speed	0.370	93%	No effect *	-	Appendix A
9.	Cadence	0.220	0%	-	-	-

* The removal of Amatachaya, Keawsutthi [60] did not change the *p*-value but led to an effect size change in the opposite direction (initial analysis: 0.76, leave-one-out: −0.02).

## Data Availability

The data can be provided upon reasonable request.

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
