# Peer review of "Does Music Therapy Improve Gait after Traumatic Brain Injury and Spinal Cord Injury? A Mini Systematic Review and Meta-Analysis"

_brainsci, 2023, doi:10.3390/brainsci13030522_

Round 1
Reviewer 1 Report
The manuscript concerns a systematic review and meta-analysis of scientific articles
on the influence of music therapy on spatiotemporal aspects of gait in people with
TBI and SCI.
The idea of a systematic review and meta-analysis as well as the summary of the
results of these studies are very interesting.
Some doubts are related, among others, to: heterogeneity of qualified studies, also reported by the authors.
Author Response
Comment: The manuscript concerns a systematic review and meta-analysis of scientific articles on the influence of music therapy on spatiotemporal aspects of gait in people with TBI and SCI.
The idea of a systematic review and meta-analysis as well as the summary of the results of these studies are very interesting. Some doubts are related, among others, to: heterogeneity of qualified studies, also reported by the authors.
Response: We sincerely thank the reviewer for taking the time to review our paper and providing valuable feedback. We appreciate your comments and agree that the heterogeneity of the qualified studies is an important consideration in our analysis. To address this concern, we have included a leave-one-out sensitivity analysis in our review. This analysis allows us to assess the influence of individual studies on the overall results, thereby providing a more robust and reliable conclusion. The changes are mentioned at:
Page 4, Line 152-156: We also conducted “leave-one-out” sensitivity analyses to test the robustness of our findings and explore the heterogeneity. The method systematically removes each study from the meta-analysis and re-analyzes the data to assess the influence of individual studies on the overall results. This helps to identify studies that may be driving the results and assess the robustness of the findings (1).
Page 19-20, Line 105-109: Sensitivity analysis
A comprehensive account of the leave-one-out sensitivity analysis is presented in Table 5. In particular, studies were reported in the table if the significance level of the global analysis was less than 0.05 and the exclusion of any individual study caused the significance level to rise above this threshold. Conversely, studies were also reported if the overall analysis was not significant at a 0.05 level, and the exclusion of any specific study led to a decrease in the significance level below this threshold.
Page 23, Line 116: Table 5
Supplementary figures S1 to S5
Reviewer 2 Report
Authors did great job in summarizing the work. I liked how paper was presented and well formulated.
Author Response
Comment: Authors did great job in summarizing the work. I liked how paper was presented and well formulated.
Response: We sincerely thank the reviewer for this comment.
Reviewer 3 Report
The manuscrpit entitled “Does Music Therapy Improve Gait After Traumatic Brain Inuries and Spinal Cord Injury? A Mini Systematic Review and Meta-analysis is very interesting, discussing the inclusion of music therapy and its impact on walking speed and stride length in patients with TBI and SCI. Rehabilitation of this group of patients is very difficult and does not always bring the desired results. The search for new methods that can improve the quality of life of patients after injuries is very much needed. Of course, music therapy cannot replace rehabilitation as the primary method of recovery for patients after SCI and TBI, but it can bring new elements.
The paper lacks information that music therapy was used in patients with SCI who had only an ASIA C and D score, i.e. they had neurological damage that allowed them to rehabilitate their gait. Patients with type B and A lesions are usually unable to move. Please let the authors include this information. Information on studies examining the impact of music therapy on cognitive functions and depressive disorders in patients after SCI and TBI would also be important.
Author Response
Comment 1: The paper lacks information that music therapy was used in patients with SCI who had only an ASIA C and D score, i.e. they had neurological damage that allowed them to rehabilitate their gait. Patients with type B and A lesions are usually unable to move. Please let the authors include this information.
Response: We thank the reviewer for this critical comment. We agree with the reviewer’s concern. Indeed, our results are relatable to SCI patients with ASIA C and D score and not ASIA A and B. We have now stated this clearly in the manuscript, the changes are mentioned in the limitations section of the manuscript at:
Page 24, Line 229-234: Moreover, the study's findings regarding the influence of MT on SCI are limited because the review only included studies with individuals classified as ASIA C or D. These individuals had neurological damage but were still able to rehabilitate their gait. In contrast, individuals with type B and A lesions, who typically cannot move, were not assessed. Therefore, it is important to interpret the study's results with caution, as they may not be applicable to the entire SCI population.
Comment 2: Information on studies examining the impact of music therapy on cognitive functions and depressive disorders in patients after SCI and TBI would also be important.
Response: Thank you for your comment, which we appreciate. However, our primary objective was to examine the impact of music therapy on spatiotemporal outcomes of gait, as stated in our pre-registration protocol on the Open Science Framework (OSF) platform (https://osf.io/crmpw). Therefore, we did not consider studies that evaluated the effect of music therapy on cognitive functions or depressive disorders in patients with SCI and TBI as they were beyond the scope of our review. We sincerely regret this inconvenience. We have acknowledged this limitation in the limitations section of our manuscript, the changes are mentioned at:
Page 24, Line 247-254: Another limitation of our review is that we did not include studies that investigated the effects of MT on cognitive and psychological outcomes in individuals with TBI and SCI, as they fell outside the scope of our research question. Although several studies have reported positive effects of MT on these outcomes (2, 3, 4), we were unable to evaluate them in our review. However, we suggest that future systematic reviews be conducted to establish the current state of evidence regarding the impact of MT on cognitive and psychological outcomes in individuals with TBI and SCI.
Round 2
Reviewer 3 Report
Dear Authors., Thank you for your replay. Your comments satisfy me.